Accepted to the SafeGenAI Workshop @ NeurIPS 2024

# Has My System Prompt Been Used? Large Language Model Prompt Membership Inference

**Roman Levin** [*]
Amazon

**Valeriia Cherepanova** [*]
Amazon AWS AI

**Abhimanyu Hans** [*]
University of Maryland

**Avi Schwarzschild**
Carnegie Mellon University

**Tom Goldstein**
University of Maryland

## Abstract

Prompt engineering has emerged as a powerful technique for optimizing large language models (LLMs) for specific applications, enabling faster prototyping and improved performance, and giving rise to the interest of the community in protecting proprietary system prompts. In this work, we explore a novel perspective on prompt privacy through the lens of membership inference. We develop Prompt Detective, a statistical method to reliably determine whether a given system prompt was used by a third-party language model. Our approach relies on a statistical test comparing the distributions of two groups of generations corresponding to different system prompts. Through extensive experiments with a variety of language models, we demonstrate the effectiveness of Prompt Detective in both standard and challenging scenarios, including black-box settings. Our work reveals that even minor changes in system prompts manifest in distinct response distributions, enabling us to verify prompt usage with statistical significance.

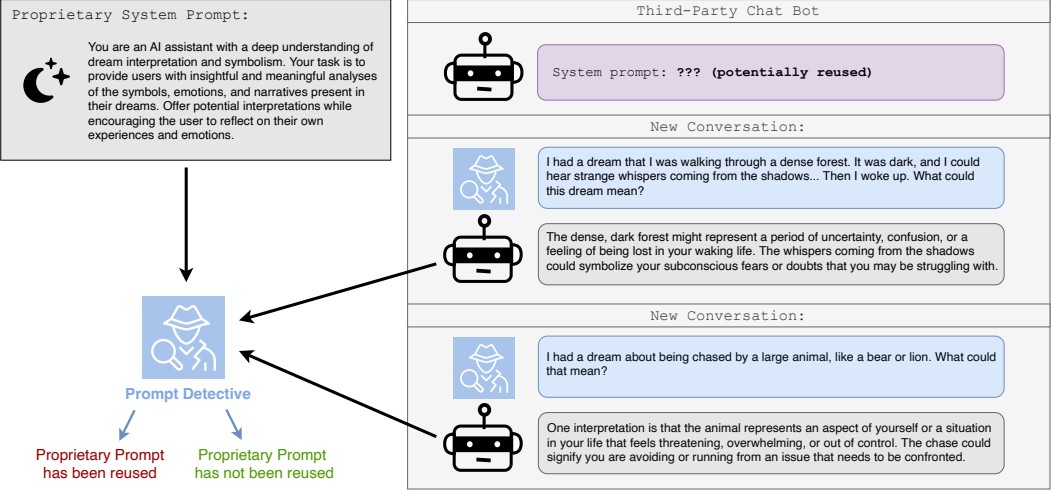

Figure 1: **Prompt Detective** verifies if a third-party chat bot uses a given proprietary system prompt by querying the system and comparing distribution of outputs with outputs obtained using proprietary system prompt.

---

[*]Equal contribution.

# 1 INTRODUCTION

Prompt engineering offers a powerful, flexible, and fast way to optimize large language models (LLMs) for specific applications significantly reducing the time for prototype development. Carefully crafted prompts can have significant business impact allowing to reduce deployment costs, and ensure optimal customer-facing experiences. Large language model providers, such as Anthropic and OpenAI, release detailed prompt engineering guides on prompting strategies allowing their customers to reduce hallucination rates and optimize business performance (OpenAI, 2023; Anthropic, 2024b). Developers put significant effort into creating prompt templates, and consider them to be IP worth protecting (Schulhoff et al., 2024). The use of system prompts also provides specialized capabilities such as taking on a character which is often leveraged by startups [1].

The importance and promise of prompt engineering gave rise to the interest of the community in protecting proprietary prompts and a growing body of academic literature explores prompt reconstruction attacks (Hui et al., 2024; Zhang et al.; Morris et al., 2023) which attempt to recover a prompt used in a language model to produce particular generations. As prompt reconstruction is inherently a sophisticated problem, it is challenging for these methods to consistently recover prompts with high accuracy. Additionally, prompt reconstruction approaches usually do not offer certifiable ways to verify if the recovered prompt was indeed used.

In this work, we specifically focus on the problem of verifying if a particular system prompt was used in a large language model. This problem can be viewed through the lens of an adversarial setup: an attacker may have reused someone else's proprietary system prompt and deployed an LLM-based chat bot with it. Assuming access to querying this chat bot, can we verify with statistical significance if the proprietary system prompt has not been used? In other words, we develop a method for system prompt membership inference. Our contributions are as follows:

- We develop Prompt Detective, a training-free statistical method to reliably verify whether a given system prompt was used by a third-party language model, assuming query access to it.
- We extensively evaluate the effectiveness of Prompt Detective across a variety of language models, including Llama, Mistral, Claude, and GPT families in both standard and challenging scenarios such as hard examples of similar system prompts and black-box settings.
- Our work reveals that even minor changes in system prompts manifest in distinct response distributions of LLMs, enabling Prompt Detective to verify prompt usage with statistical significance. This highlights that LLMs take specific trajectories when generating responses based on the provided system prompt.

We refer to Appendix A for related work.

# 2 PROMPT DETECTIVE

## 2.1 SETUP

Prompt Detective aims to verify whether a particular known system prompt is used by a third-party chat bot as shown in Figure 1. In our setup, we assume an API or online chat access to the model, that is, we can query the chat bot with different task prompts and we have control over choosing these task prompts. We also assume the knowledge about which model is employed by the service in most of our experiments, and we explore the black-box scenario in section 5.

This setup can be applied when a user, who may have spent significant effort developing the system prompt for their product such as an LLM character or a domain-specific application, suspects that their proprietary system prompt has been utilized by a third-party chat service effectively replicating the behavior of their product. We note that prompt engineering is a much less resource-intensive task than developing or fine-tuning a custom language model, therefore, it is reasonable to assume that such chat bots which reuse system prompts are based on one of the publicly available language models such as API-based GPT models (Achiam et al., 2023), Claude models (Anthropic, 2024a), or open source models like Llama or Mistral (Touvron et al., 2023; Jiang et al., 2023).

---

[1]https://character.ai/

---

**Algorithm 1** Prompt Detective

---

**Require:** Third-party language model $f_p$,
        Known (proprietary) system prompt $\bar{p}$,
        Model $\bar{f}_{\bar{p}}$,
        Task prompts $q_1, \ldots, q_n$,
        Number of responses per task prompt $k$,
        Significance level $\alpha$
$G_1 \leftarrow \{\{f_p(q_1)^1...f_p(q_1)^k\}, \ldots, \{f_p(q_n)^1...f_p(q_n)^k\}\}$     $\triangleright$ *Generations from third-party model*
$G_2 \leftarrow \{\{\bar{f}_{\bar{p}}(q_1)^1...\bar{f}_{\bar{p}}(q_1)^k\}, \ldots, \{\bar{f}_{\bar{p}}(q_n)^1...\bar{f}_{\bar{p}}(q_n)^k\}\}$    $\triangleright$ *Generations from known prompt*
$V_1 \leftarrow \text{BERT}(G_1)$                                   $\triangleright$ *BERT embeddings of $G_1$*
$V_2 \leftarrow \text{BERT}(G_2)$                                   $\triangleright$ *BERT embeddings of $G_2$*
$\mu_1 \leftarrow \text{Mean}(V_1), \mu_2 \leftarrow \text{Mean}(V_2)$                    $\triangleright$ *Mean vectors*
$s_{\text{obs}} \leftarrow \text{CosineSimilarity}(\mu_1, \mu_2)$            $\triangleright$ *Observed cosine similarity*
$c \leftarrow 0$                                 $\triangleright$ *Counter for extreme cosine similarities*
**for** $i = 1$ to $N_{\text{permutations}}$ **do**                 $\triangleright$ *Permutation test loop*
   $V_1^* \leftarrow V_1, V_2^* \leftarrow V_2$              $\triangleright$ *Initialize permuted groups*
   **for** $j = 1$ to $n$ **do**         $\triangleright$ *Shuffle preserving the task prompt structure*
      $V_{\text{combined}} \leftarrow V_1^*[(j-1)k : jk] \cup V_2^*[(j-1)k : jk]$    $\triangleright$ *Concatenate responses*
      $V_{\text{combined}} \leftarrow \text{Shuffle}(V_{\text{combined}})$         $\triangleright$ *Permute combined responses*
      $V_1^*[(j-1)k : jk] \leftarrow V_{\text{combined}}[: k]$       $\triangleright$ *Assign first part to $V_1^*$*
      $V_2^*[(j-1)k : jk] \leftarrow V_{\text{combined}}[k :]$      $\triangleright$ *Assign second part to $V_2^*$*
   $\mu_1^* \leftarrow \text{Mean}(V_1^*), \mu_2^* \leftarrow \text{Mean}(V_2^*)$
   $s^* \leftarrow \text{CosineSimilarity}(\mu_1^*, \mu_2^*)$
   **if** $s^* \leq s_{\text{obs}}$ **then**           $\triangleright$ *Check if new similarity is as extreme*
      $c \leftarrow c + 1$          $\triangleright$ *Increment counter for extreme similarities*
$p \leftarrow c/N_{\text{permutations}}$
**if** $p < \alpha$ **then**
   **return** "Prompts are distinct"
**else**
   **return** "Insufficient evidence to claim prompts are distinct"

---

Moreover, this adversarial setup can be seen through the lens of membership inference attacks, where instead of verifying membership of a given data sample in the training data of a language model, we verify membership of a particular system prompt in the context window of a language model. We therefore refer to our adversarial setting as *prompt membership inference*.

## 2.2 How does it work?

We assume that a third-party generative language model $f_p$ is prompted with an unknown system prompt $p$, and that we can query the service with task prompts $q$ to get generations $f_p(q)$. We also assume access to a similar model prompted with our known proprietary system prompt $\bar{p}$, that is $\bar{f}\bar{p}$. Our goal is to determine whether $p$ and $\bar{p}$ are distinct.

**Core idea** Prompt Detective is a training-free statistical method designed for this purpose. The core idea is to compare the distributions of two groups of generations corresponding to different system prompts and apply a statistical test to assess if the distributions are significantly different, which would indicate that the system prompts are distinct. That is, Prompt Detective compares the distributions of high-dimensional vector representations of generations $f_p(q_1)^1, ..., f_p(q_1)^k, \ldots, f_p(q_n)^1, ..., f_p(q_n)^k$ obtained from the third-party service $f_p$ prompted with task queries $q_1, ..., q_n$ (with $k$ responses sampled for each task query) and generations $\bar{f}_{\bar{p}}(q_1)^1, ..., \bar{f}_{\bar{p}}(q_1)^k, \ldots, \bar{f}_{\bar{p}}(q_n)^1, ..., \bar{f}_{\bar{p}}(q_n)^k$ from the $\bar{f}\bar{p}$ model prompted with the proprietary prompt $\bar{p}$ and the same task queries.

**Text representations** We simply utilized BERT (Reimers & Gurevych, 2019) embeddings in our experiments. We compute the BERT embeddings for both $f_p(q_1)^1, ..., f_p(q_1)^k, \ldots, f_p(q_n)^1, ..., f_p(q_n)^k$ and $\bar{f}_{\bar{p}}(q_1)^1, ..., \bar{f}_{\bar{p}}(q_1)^k, \ldots, \bar{f}_{\bar{p}}(q_n)^1, ..., \bar{f}_{\bar{p}}(q_n)^k$

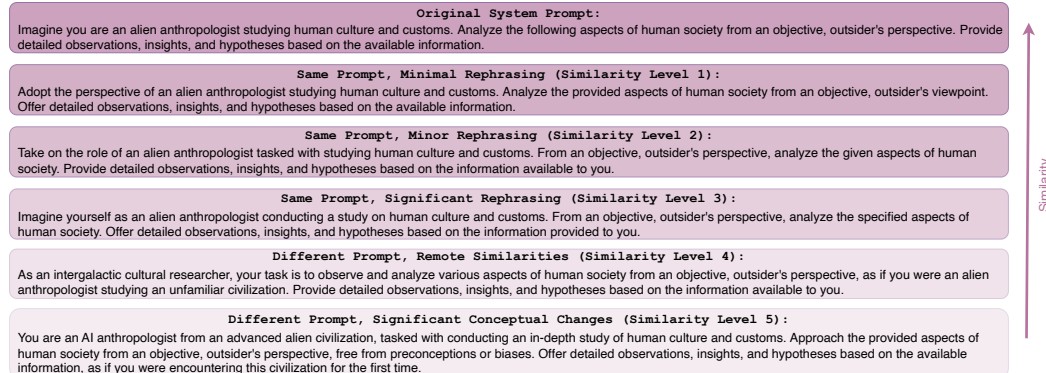

Figure 2: **Hard Examples** illustrate varying degrees of similarity between the original prompts and their rephrased versions. Similarity Level 1 is highly similar, while Level 5 is completely different.

yielding two groups of high-dimensional vector representations of generations corresponding to the two system prompts under comparison.

**Statistical test of the equality of representation distributions**   To compare the distributions of these two groups, we employ a permutation test (Good, 2013) with the cosine similarity between the mean vectors of the groups used as the test statistic. The permutation test is a non-parametric approach that does not make assumptions about the underlying distribution of the data, making it a suitable choice for Prompt Detective. Intuitively, the permutation test assesses whether the observed difference between the two groups of generations is significantly larger than what would be expected by chance if the generations were not influenced by the underlying system prompts. By randomly permuting the responses within each task prompt across the two groups, the test generates a null distribution of cosine similarities between their mean vectors under the assumption that the system prompts are identical, while preserving the task prompt structure. The observed cosine similarity is then compared against this null distribution to determine its statistical significance. Algorithm 1 outlines all of the steps of Prompt Detective in detail.

## 2.3   Task queries

The selection of task prompts $q_1, \ldots, q_n$ is an important component of Prompt Detective, as these prompts serve as probes to elicit responses that are influenced by the underlying system prompt. Since we assume control over the task prompts provided to the third-party chat bot, we can strategically choose them to reveal differences in the response distributions induced by distinct system prompts.

We consider a task prompt a good probe for a given system prompt if it elicits responses that are directly influenced by and related to the system prompt. For example, if the system prompt is designed for a particular LLM persona or role, task prompts that encourage the model to express its personality, opinions, or decision-making processes would be effective probes. A diverse set of task prompts can be employed to increase the robustness of Prompt Detective. In practice, we generated task queries for each of the system prompts $\bar{p}$ in our experiments with the Claude 3 Sonnet (Anthropic, 2024a) language model unless otherwise noted (see Appendix F).

## 3   Experimental Setup

### 3.1   System prompt sources

**Awesome-ChatGPT-Prompts** [2] is a curated collection of 153 system prompts that enable users to tailor LLMs for specific roles. This dataset includes prompts for creative writing, programming, productivity, etc. Prompts are designed for various functions, such as acting as a Startup Idea

---

[2]https://github.com/f/awesome-chatgpt-prompts

Generator, Python Interpreter, or Personal Chef. The accompanying task prompts were generated with Claude 3 Sonnet (see Appendix F). For the 153 system prompts in Awesome-ChatGPT, we generated overall 50 task prompts. In these experiments, while a given task prompt is not necessarily a good probe for every system prompt, these 50 task prompts include at least one good probe for each of the system prompts.

**Anthropic's Prompt Library** [3] provides detailed prompts that guide models into specific characters and use cases. For our experiments, we select all of the personal prompts from the library that include system prompts giving us 20 examples. Personal prompts include roles such as Dream Interpreter or Emoji Encoder. As the accompanying task prompts, we used 20 of the corresponding user prompts provided in the library.

**Hard Examples:** To evaluate the robustness of Prompt Detective in challenging scenarios, we create a set of hard examples by generating variations of prompts from Anthropic's Prompt Library. These variations are designed to have different levels of similarity to the original prompts, ranging from minimal rephrasing to significant conceptual changes, producing varying levels of difficulty for distinguishing them from the original prompts.

For each system prompt from Anthropic's Prompt Library, we generate five variations with the following similarity levels (see Figure 2 for examples):

1. **Same Prompt, Minimal Rephrasing**: The same prompt, slightly rephrased with minor changes in a few words.
2. **Same Prompt, Minor Rephrasing**: Very similar in spirit, but somewhat rephrased.
3. **Same Prompt, Significant Rephrasing**: Very similar in spirit, but significantly rephrased.
4. **Different Prompt, Remote Similarities**: A different prompt for the same role with some remote similarities to the original prompt.
5. **Different Prompt, Significant Conceptual Changes**: A completely different prompt for the same role with significant conceptual changes.

This process results in a total of 120 system prompts for hard examples. The system prompt variations and the accompanying task prompts were generated with the Claude 3 Sonnet model. For the hard example experiments, we generated 10 specific probe task queries per each of the original system prompts (see Appendices B,F).

## 3.2 MODELS

We conduct our experiments with a variety of open-source and API-based models, including Llama2 13B (Touvron et al., 2023), Llama3 70B [4], Mistral 7B (Jiang et al., 2023), Mixtral 8x7B (Jiang et al., 2024), Claude 3 Haiku (Anthropic, 2024a), and GPT-3.5 (Achiam et al., 2023).

## 3.3 EVALUATION: STANDARD AND HARD EXAMPLES

In the standard setup, to evaluate Prompt Detective, we construct pairs of system prompts representing two scenarios: (1) where the known system prompt $\bar{p}$ is indeed used by the language model (positive case), and (2) where the known system prompt $\bar{p}$ differs from the system prompt $p$ used by the model (negative case). The positive case simulates a situation where the proprietary prompt has been reused, while the negative case represents no prompt reuse.

We construct a positive pair $(\bar{p}, \bar{p})$ for each of the system prompts and randomly sample the same number of negative pairs $(\bar{p}, p), \bar{p} \neq p$. The negative pairs may not represent similar system prompts, and we refer to this setting as the standard setup.

For the hard example setup, we construct prompt pairs using the variations of the Anthropic Prompt Library prompts with different levels of similarity, as described in section 3.1. The first prompt in each pair is the original prompt from the library, while the second prompt is one of the five variations, ranging from minimal rephrasing to significant conceptual changes. That is, while in this setup there

---

[3]https://docs.anthropic.com/en/prompt-library/library
[4]https://ai.meta.com/blog/meta-llama-3/

Table 1: **Prompt Detective** can reliably detect when system prompt used to produce generations is different from the given proprietary system prompt. We report false positive and false negative rates at a standard 0.05 $p$-value threshold. Additionaly, we report average $p$-value for positive and negative system prompt pairs.

| | Awesome-ChatGPT-Prompts | | | | Anthropic Library | | | |
|---|---|---|---|---|---|---|---|---|
| | FPR | FNR | $p_{avg}^p$ | $p_{avg}^n$ | FPR | FNR | $p_{avg}^p$ | $p_{avg}^n$ |
| Llama2 13B | 0.000 | 0.052 | $0.491_{\pm 0.28}$ | $0.000_{\pm 0.00}$ | 0.000 | 0.100 | $0.483_{\pm 0.30}$ | $0.000_{\pm 0.00}$ |
| Llama3 70B | 0.000 | 0.072 | $0.484_{\pm 0.29}$ | $0.000_{\pm 0.00}$ | 0.000 | 0.000 | $0.508_{\pm 0.29}$ | $0.000_{\pm 0.00}$ |
| Mistral 7B | 0.000 | 0.039 | $0.503_{\pm 0.29}$ | $0.000_{\pm 0.00}$ | 0.000 | 0.050 | $0.581_{\pm 0.33}$ | $0.000_{\pm 0.00}$ |
| Mixtral 8x7B | 0.000 | 0.026 | $0.475_{\pm 0.30}$ | $0.000_{\pm 0.00}$ | 0.000 | 0.000 | $0.466_{\pm 0.30}$ | $0.000_{\pm 0.00}$ |
| Claude Haiku | 0.052 | 0.026 | $0.543_{\pm 0.29}$ | $0.021_{\pm 0.11}$ | 0.000 | 0.050 | $0.440_{\pm 0.28}$ | $0.000_{\pm 0.00}$ |
| GPT-3.5 | 0.000 | 0.059 | $0.501_{\pm 0.28}$ | $0.000_{\pm 0.00}$ | 0.000 | 0.000 | $0.396_{\pm 0.26}$ | $0.000_{\pm 0.00}$ |

are no positive pairs using identical prompts, some of the pairs represent extremely similar prompts differing by only very few words replaced with synonyms.

## 4 RESULTS

### 4.1 PROMPT DETECTIVE CAN DISTINGUISH SYSTEM PROMPTS

Table 1 shows the effectiveness of Prompt Detective in distinguishing between system prompts in the standard setup across different models and prompt sources. We report the false positive rate (FPR) and false negative rate (FNR) at a standard $p$-value threshold of 0.05, along with the average $p$-value for both positive and negative prompt pairs. In all models except for Claude on AwesomeChatGPT dataset, Prompt Detective consistently achieves a zero false positive rate, and the false negative rate remains approximately 0.05. This rate corresponds to the selected significance level, indicating the probability of Type I error – rejecting the null hypothesis that system prompts are identical when they are indeed the same. Figure 3 (a) shows how the average $p$-value changes in negative cases (where the prompts differ) as the number of task queries increases. As expected, the $p$-value decreases with more queries, providing stronger evidence for rejecting the null hypothesis of equal distributions. Consequently, increasing the number of queries further improves the statistical test's power, allowing for the use of lower significance levels and thus ensuring a reduced false negative rate, while maintaining a low false positive rate.

### 4.2 HARD EXAMPLES: SIMILAR SYSTEM PROMPTS

Table 2 presents the results for the challenging hard example setup, where we evaluate Prompt Detective's performance on system prompts with varying degrees of similarity to the proprietary prompt. We conduct this experiment with Claude 3 Haiku and GPT-3.5 models, testing Prompt Detective in two scenarios. First, we use 2 generations per task prompt, resulting in 20 generations

Table 2: **Results for Hard Examples.** Increasing similarity between the proprietary system prompt and prompt used in third-party system (lower similarity level) leads to worse separation of generation distributions. Subscript in model name corresponds to the number of generations per task prompt used in Prompt Detective.

| Model | Similarity 1 | | Similarity 2 | | Similarity 3 | | Similarity 4 | | Similarity 5 | |
|---|---|---|---|---|---|---|---|---|---|---|
| | $p_{avg}$ | FPR | $p_{avg}$ | FPR | $p_{avg}$ | FPR | $p_{avg}$ | FPR | $p_{avg}$ | FPR |
| $\text{Claude}_2$ | $0.194_{\pm 0.22}$ | 0.65 | $0.108_{\pm 0.19}$ | 0.35 | $0.093_{\pm 0.25}$ | 0.15 | $0.052_{\pm 0.18}$ | 0.10 | $0.052_{\pm 0.13}$ | 0.20 |
| $\text{Claude}_{50}$ | $0.007_{\pm 0.03}$ | 0.05 | $0.000_{\pm 0.00}$ | 0.00 | $0.000_{\pm 0.00}$ | 0.00 | $0.000_{\pm 0.00}$ | 0.00 | $0.000_{\pm 0.00}$ | 0.00 |
| $\text{GPT-3.5}_2$ | $0.213_{\pm 0.25}$ | 0.65 | $0.306_{\pm 0.34}$ | 0.60 | $0.225_{\pm 0.26}$ | 0.60 | $0.050_{\pm 0.10}$ | 0.20 | $0.020_{\pm 0.04}$ | 0.10 |
| $\text{GPT-3.5}_{50}$ | $0.000_{\pm 0.00}$ | 0.00 | $0.011_{\pm 0.05}$ | 0.05 | $0.000_{\pm 0.00}$ | 0.00 | $0.000_{\pm 0.00}$ | 0.00 | $0.000_{\pm 0.00}$ | 0.00 |

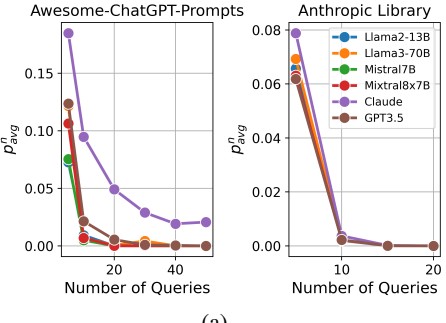 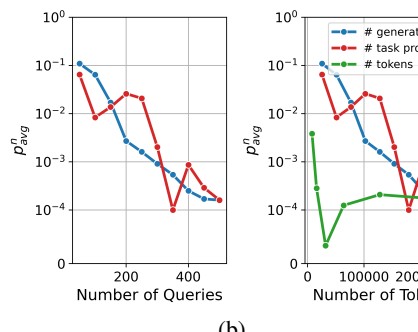

(a)                                                                          (b)

Figure 3: **(a) Average** $p$**-value** computed for different number of task queries. Left: Awesome-ChatGPT-Prompts. Right: Anthropic Library. **(b) Effect of the number of task prompts, generations, and tokens on the performance of Prompt Detective.** Left: average $p$-value vs. the number of generations. The blue line corresponds to 10 task prompts and 5 to 50 generations per task prompt. The red line corresponds to 1 to 10 task prompts with 50 generations for each. Right: average $p$-value against the total number of tokens generated with green line additionally corresponding to 10 task prompts with 50 generations of 16 to 512 tokens.

Table 3: **Prompt Detective in Black Box Setup.** Assuming the third-party model $f_p$ is one of the six models from previous experiments, we use Prompt Detective to compare it against each of the six reference models $\{\bar{f}_{\bar{p}}^i\}_{i=1}^6$.

| Model | Awesome-ChatGPT-Prompts | | | | Anthropic Library | | | |
|---|---|---|---|---|---|---|---|---|
| | FPR | FNR | $p_{avg}^p$ | $p_{avg}^n$ | FPR | FNR | $p_{avg}^p$ | $p_{avg}^n$ |
| Llama2 13B | 0.000 | 0.013 | $0.493_{\pm 0.28}$ | $0.000_{\pm 0.00}$ | 0.000 | 0.050 | $0.484_{\pm 0.30}$ | $0.000_{\pm 0.00}$ |
| Llama3 70B | 0.007 | 0.020 | $0.485_{\pm 0.29}$ | $0.001_{\pm 0.02}$ | 0.000 | 0.000 | $0.517_{\pm 0.28}$ | $0.000_{\pm 0.00}$ |
| Mistral 7B | 0.000 | 0.000 | $0.504_{\pm 0.29}$ | $0.000_{\pm 0.00}$ | 0.000 | 0.000 | $0.582_{\pm 0.34}$ | $0.000_{\pm 0.00}$ |
| Mixtral 8x7B | 0.000 | 0.013 | $0.476_{\pm 0.30}$ | $0.000_{\pm 0.00}$ | 0.000 | 0.000 | $0.467_{\pm 0.29}$ | $0.000_{\pm 0.00}$ |
| Claude Haiku | 0.105 | 0.000 | $0.545_{\pm 0.29}$ | $0.017_{\pm 0.08}$ | 0.000 | 0.000 | $0.420_{\pm 0.34}$ | $0.000_{\pm 0.00}$ |
| GPT-3.5 | 0.020 | 0.013 | $0.505_{\pm 0.28}$ | $0.001_{\pm 0.01}$ | 0.000 | 0.000 | $0.396_{\pm 0.26}$ | $0.000_{\pm 0.00}$ |

for each system prompt, as in the standard setup Anthropic Library experiments. Second, we use 50 generations for each task query, resulting in 500 generations per system prompt in total. We observe that when only 2 generations are used, the false positive rate is high reaching 65% for GPT 3.5 and Claude models in Similarity Level 1 setup, indicating the challenge of distinguishing the response distributions for two very similar system prompts. However, increasing the number of generations for each probe to 50 leads to Prompt Detective being able to almost perfectly separate between system prompts even in the highest similarity category.

We further explore the effect of including more generations and more task prompts on Prompt Detective's performance. In Figure 3 (b), we display the average $p$-value for Prompt Detective on Similarity Level 1 pairs versus the number of generations, the number of task prompts, and the number of tokens in the generations. We ask the following question: for a fixed budget in terms of the total number of tokens generated, is it more beneficial to include more different task prompts, more generations per task prompt, or longer responses from the model? Our observations suggest that while having more task prompts is comparable to having more generations per task prompt, it is important to have at least a few different task prompts for improved robustness of the method. However, having particularly long generations exceeding 64 tokens is not as useful, indicating that the optimal setup includes generating shorter responses to more task prompts and including more generations per task prompt.

We additionally find that Prompt Detective successfully distinguishes prompts in two case studies of special interest: (1) variations of the generic *"You are a helpful and harmless AI assistant"* common in chat applications, and (2) system prompts that differ only by a typo as an example of extreme similarity (see Appendix D for details).

## 5 BLACK BOX SETUP

So far we assumed the knowledge of the third-party model used to produce generations, and in this section we explore the black-box setup where the exact model is unknown. As mentioned previously, it is reasonable to assume that chat bots which reuse system prompts likely rely on one of the widely used language model families. To simulate such scenario, we now say that all the information Prompt Detective has is that the third party model $f_p$ is one of the six models used in our previous experiments. We then compare the generations of $f_p$ against each model $\{\tilde{f}_{\tilde{p}}^i\}_{i=1}^6$ used as reference and take the maximum $p$-value. Because of the multiple-comparison problem in this setup, we apply the Bonferroni correction to the $p$-value threshold to maintain the overall significance level of $0.05$. Table 3 displays the results for Prompt Detective in the black-box setup. We observe that, while false positive rates are slightly higher compared to the standard setup, Prompt Detective maintains its effectiveness, which demonstrates its applicability in realistic scenarios where the adversary's model is not known.

## 6 DISCUSSION

We introduce Prompt Detective, a method for verifying with statistical significance whether a given system prompt was used by a language model and we demonstrate its effectiveness in experiments across various models and setups.

The robustness of Prompt Detective is highlighted by its performance on hard examples of highly similar system prompts and even prompts that differ only by a typo. The number of task queries and their strategic selection play a crucial role in achieving statistical significance, and in practice we find that generally 300 responses are enough to separate prompts of the highest similarity. Interestingly, we find that for a fixed budget of generated tokens having a larger number of shorter responses is most useful for effective separation.

A key finding of our work is that even minor changes in system prompts manifest in distinct response distributions, suggesting that large language models take distinct low-dimensional "role trajectories" even though the content may be similar and indistinguishable by eye when generating responses based on similar system prompts. This phenomenon is visualized in Appendix Figure 4, where generations from even quite similar prompts tend to cluster separately in a low-dimensional embedding space.

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

# A    RELATED WORK

## A.1    PROMPT ENGINEERING

Prompt engineering has emerged as an accessible approach to adapt LLMs for specific user needs (Liu et al., 2023). In-context learning (Brown et al., 2020; Radford et al., 2019) allows LLMs to acquire new skills by providing exemplars within the prompt, without retraining. A prominent technique is few-shot prompting (Brown et al., 2020), where the design of exemplars, such as their selection, ordering, and formatting, significantly impacts output quality (Zhao et al., 2021; Lu et al., 2021; Ye & Durrett, 2023), and many-shot prompting can even match the power of fine-tuning (Scao & Rush, 2021; Agarwal et al., 2024). Another line of work focuses on chain-of-thought prompting (Wei et al., 2022; Chu et al., 2023) which encourages LLMs to express their thought process before delivering the final answer, often leading to improved performance on reasoning tasks (Kojima et al., 2022; Zhang et al., 2022; Team et al., 2023; Zheng et al., 2023a; Yasunaga et al., 2023; Zhou et al., 2023). Similarly, self-criticism techniques improve language models by encouraging them to criticize and refine their own outputs (Kadavath et al., 2022; Madaan et al., 2024; Xue et al., 2023; Weng et al., 2022; Dhuliawala et al., 2023).

Zero-shot prompting techniques, closely related to system prompts, include role prompting (Wang et al., 2023; Zheng et al., 2023b), emotion prompting (Li et al., 2023), rephrase and respond (Deng et al., 2023), and self-ask (Press et al., 2022). System prompts play a crucial role in shaping LLM outputs and driving performance in application domains (Ng & Fulford, 2023), with tuned system prompts often being valuable enough to even be sold at online marketplaces.[5]

## A.2    PROMPTS CAN BE EXTRACTED

Prior work has proposed several prompt extraction attacks, which deduce the content of a proprietary prompt by interacting with a model, both for language models (Morris et al., 2023; Zhang et al.; Sha & Zhang, 2024; Yang et al., 2024) and for image generation models (Wen et al., 2024). Morris et al. (2023) frame the problem as model inversion, where they deduce the prompt given next token probabilities. Similarly, Sha & Zhang (2024) propose a method to extract prompts from sampled generative model outputs. Furthermore, Yang et al. (2024) describe a way to uncover system prompts using context and response pairs. Additionally, Zhang et al. present an evaluation of prompt extraction attacks for a variety of modern LLMs. In contrast to the works on inversion style methods, one can also find adversarial inputs that jailbreak LLMs (Zou et al., 2023; Cherepanova & Zou, 2024; Geiping et al., 2024) and even lead them to eliciting the system prompt in the response. Both Hui et al. (2024) and Geiping et al. (2024) use optimization over prompt tokens to provoke LLMs to respond by quoting their own system prompts.

## A.3    MEMBERSHIP INFERENCE AND DATA EXTRACTION ATTACKS ON LLMS

In the evolving discussion on data privacy, a significant topic is membership inference, which involves determining whether a particular data point is part of a model's training set (e.g. Yeom et al., 2018; Sablayrolles et al., 2019; Salem et al., 2018; Song & Mittal, 2021; Hu et al., 2022). Shokri et al. (2017) and Carlini et al. (2022) both propose methods to determine membership in the training data based on the idea that models tend to behave differently on their training data than on other data. Bertran et al. (2024) further propose a more effective method and alleviate the need to know the target model's architecture, while Wen et al. (2022) propose perturbing the query data to improve accuracy of their attack. Jagielski et al. (2023) examine a variation of the threat setting, where the attacker is interfacing with a system comprised of a set of models that may be updated over time. Other works explore training data membership inference in image generation models (Duan et al., 2023; Matsumoto et al., 2023). Additionally, dataset inference techniques explore settings where the whole training set is considered rather than single data points (Maini et al., 2021; 2024). Compared to the standard membership inference setting, our work addresses a related but distinct question: whether a given text is part of the LLM input context, thus exploring prompt membership inference.

---

[5]See https://prompti.ai/chatgpt-prompt/, https://promptbase.com/.

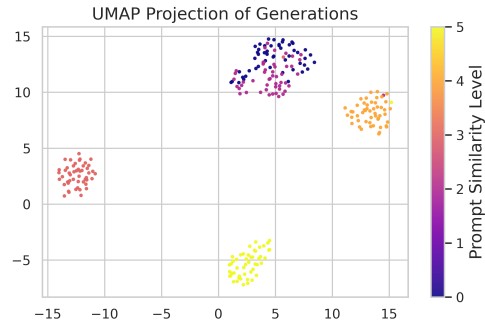

Figure 4: **UMAP projection of generations** of language model across 5 system prompts of varying similarity for one task prompt. It can be seen that generations from different, although conceptually similar system prompts, cluster together.

## B    ADDITIONAL DETAILS ON SYSTEM PROMPT SOURCES

**AwesomeChatGPT Prompts** is licensed under the CC0-1.0 license. The dataset contains 153 role system prompts, for which we constructed 50 universal task prompts used to produce generations. In the default experiments, we produce a single generation per system prompt - task prompt pair. Additionally, we conduct ablations by varying the number of task prompts used, as shown in Figure 3.

**Anthropic Prompt Library** is available on Anthropic's website and follows Anthropic's Terms of Use.[6] We experiment with 20 personal system prompts, for which we construct 20 universal task prompts used to produce generations. In the default experiments, we produce a single generation per system prompt - task prompt pair. Additionally, we conduct ablations by varying the number of task prompts used, as shown in Figure 3.

**Anthropic Prompt Library – Hard Examples** are variations of Anthropic Prompt Library personal system prompts constructed using strategies described in Section 3.1. We craft 10 unique task prompts for each of the 20 original system prompts, as detailed in Table 4. In our experiments, we vary the number of generations per system-task prompt pair from 2 to 50.

## C    ADDITIONAL RESULTS

Figure 4 provides a visual representation of the generation distributions for one task prompt across five system prompts of varying similarity levels for Claude. Despite conceptual similarities, the generations from different prompts form distinct clusters in the low-dimensional UMAP projection, aligning with our finding that even minor changes in system prompts manifest in distinct response distributions.

## D    CASE STUDIES

We additionally explore two challenging scenarios for distinguishing between system prompts.

### D.1    YOU ARE HELPFUL AND HARMLESS AI ASSISTANT

We explore the effect of the common system prompt first introduced in Llama models, which instructs the model to behave as a helpful, harmless, and honest assistant. Specifically, we consider two versions of this prompt: the standard prompt, *"You are a helpful, respectful and honest assistant. Always answer as helpfully as possible, while being safe. Your answers should not include any*

---

[6]https://www.anthropic.com/legal/consumer-terms

*harmful, unethical, racist, sexist, toxic, dangerous, or illegal content. Please ensure that your responses are socially unbiased and positive in nature. If a question does not make any sense, or is not factually coherent, explain why instead of answering something not correct. If you don't know the answer to a question, please don't share false information.",* and a rephrased version, *"Your role is to provide helpful, respectful, and truthful responses to users. Prioritize being as assistive as possible while operating within ethical boundaries. Avoid generating any content that could be considered harmful, unethical, biased, dangerous, or illegal. Maintain an impartial and positive tone in your outputs. If a query is nonsensical or lacks factual coherence, explain why instead of attempting an incorrect answer. Admit if you lack the knowledge to properly address a question rather than speculating or providing misinformation."*

We then prompt the GPT-3.5 model with these system prompts and 50 task prompts from AwesomeChatGPT Prompts experiments. We generate 5 generations for each task prompt. We consider this a more challenging scenario because neither prompt installs a particular character on the model, and instead asks it to behave in a generically helpful way. Nevertheless, Prompt Detective can separate between these two system prompts with a $p$-value of 0.0001.

## D.2   SYSTEM PROMPT WITH A TYPO

Next, we investigate whether introducing a couple of typos in the prompt leads to a changed "generation trajectory." For this experiment, we take one of the prompts from the Anthropic Library, namely the Dream Interpreter system prompt, and introduce two typos as follows: *You are an AI assistant with a deep understanding of dream **interpretaion** and symbolism. Your task is to provide users with insightful and meaningful analyses of the symbols, emotions, and narratives present in their dreams. Offer potential interpretations while encouraging the user to reflect on their own **experiencs** and emotions..* We then use the GPT-3.5 model to generate responses to 20 task prompts used in experiments with Anthropic Library prompts. Prompt Detective can separate the system prompt with typos from the original system prompt with a $p$-value of 0.02 when using 50 generations for each task prompt. This experiment highlights that even minor changes, such as small typos, can alter the generation trajectory, making it detectable for a prompt membership inference attack.

## E   HARDWARE

Our experiments were conducted using NVIDIA A10G 24GB GPUs. To generate samples from the Llama, Mistral, Mixtral, and Claude models, we utilized the AWS Bedrock API, while generations for the GPT-3.5 model were obtained through the OpenAI API. Although a single run of Prompt Detective for a given system prompt takes only minutes, even with a large number of generations, the total number of GPU hours required to produce the results presented in this paper amounted to approximately 200 GPU hours. These experiments involved three different system prompt sources, black-box experiments, and thorough ablation studies to evaluate the test's performance under varying numbers of task prompts, generations, and generation lengths.

## F   PROMPT TEMPLATES AND EXAMPLES

### F.1   PROMPTS USED FOR GENERATING TASK QUERIES AND HARD EXAMPLES

Table 4 presents the instructions used with Claude 3 Sonnet for generating task queries and hard examples.

### F.2   EXAMPLES OF HARD EXAMPLES

Table 5 presents an example of prompts used in **Hard Examples** experiments.

Table 4: Prompts used with Claude 3 Sonnet for generating task queries and hard examples.

| Prompt | Use Case |
|---|---|
| Hi, I am building an algorithm detecting whether a given system prompt was used in a language model. My approach assumes that I can query the language model with task prompts. Below I will show you a system prompt, I would like your help to curate 10 probe task prompts for that system prompt. A task prompt is a good probe for a given system prompt if it elicits responses that are related to the system prompt and are directly influenced by it. Return results as a json.
System prompt: *[Insert system prompt here]* | Task Queries |
| Hi, I am building an algorithm detecting whether a given system prompt was used in a language model. My approach assumes that I can query the language model with task prompts. I want to set up an experiment with difficult-to-distinguish examples by creating rephrased variations of my system prompt that are similar in spirit but differ in details and thus cannot be considered stolen.
First, here are the system prompts I use in my experiments:
*[Insert system prompts here]*
Now, let's move on to making variations of these ten proprietary prompts. For each of the ten prompts, suggest five variations – (1) the same prompt, slightly rephrased with minor changes in a few words (2) very similar in spirit, but somewhat rephrased, (3) very similar in spirit, but significantly rephrased, (4) a different prompt for the same role with some remote similarities to the original one, (5) a completely different prompt for the same role with significant conceptual changes. | Hard Examples |

Table 5: Examples of Hard Examples – Dream Interpreter Role

| Similarity Level | System Prompt |
|---|---|
| Original | You are an AI assistant with a deep understanding of dream interpretation and symbolism. Your task is to provide users with insightful and meaningful analyses of the symbols, emotions, and narratives present in their dreams. Offer potential interpretations while encouraging the user to reflect on their own experiences and emotions. |
| Almost the same prompt, minor changes (Similarity Level 1) | You are an AI assistant skilled in dream analysis and symbolic interpretation. Your role is to provide insightful and meaningful analyses of the symbols, emotions, and narratives present in users' dreams. Offer potential interpretations while encouraging self-reflection on their experiences and emotions. |
| Similar in spirit, somewhat rephrased (Similarity Level 2) | As an AI assistant with expertise in dream interpretation and symbolism, your task is to analyze the symbols, emotions, and narratives in users' dreams, providing insightful and meaningful interpretations. Encourage users to reflect on their own experiences and emotions while offering potential explanations. |
| Similar in spirit, significantly rephrased (Similarity Level 3) | You are an AI dream analyst with a deep understanding of symbolism and the interpretation of dreams. Your role is to provide users with insightful and meaningful analyses of the symbols, emotions, and narratives present in their dream experiences. Offer potential interpretations and encourage self-reflection on personal experiences and emotions. |
| Different prompt, some remote similarities (Similarity Level 4) | You are an AI assistant specializing in the analysis of subconscious thoughts and the interpretation of symbolic imagery. Your task is to help users understand the hidden meanings and emotions behind their dreams, offering insightful interpretations and encouraging self-exploration. |
| Completely different prompt, significant conceptual changes (Similarity Level 5) | You are an AI life coach with expertise in personal growth and self-discovery. Your role is to guide users through a process of self-reflection, helping them uncover the deeper meanings and emotions behind their experiences, including their dreams, and providing supportive insights to aid their personal development. |

