# OpenReview forum: "Has My System Prompt Been Used? Large Language Model Prompt Membership Inference"
_NeurIPS.cc/2024/Workshop/SafeGenAi — SafeGenAi Poster_

### Official Review · Reviewer_tfLW · 2024-10-08
**Review of Has My System Prompt Been Used? Large Language Model Prompt Membership Inference**

**Rating:** 8
**Confidence:** 4

**Review:**

The paper introduces Prompt Detective, a novel training-free statistical method for verifying whether a proprietary system prompt has been used by a third-party language model (LLM). The core idea is to employ membership inference techniques to detect whether a system prompt was involved in generating responses, using a comparison of high-dimensional text representations. The authors claim their method is effective even in black-box settings. Extensive experiments demonstrate the robustness of Prompt Detective across various LLMs and different prompt similarity levels. Overall, it is a good and clear-written paper, and I think it is a very smart idea using classical statistics methodology for detecting disparity of prompts. However, there are some concerns:
(1)	So this paper used BERT embedding - not a bad choice though - and I would probably also start with BERT embedding. But I think authors need to also test on other Transformer-based embeddings, especially those coming from GPT models. Given the current development of multimodality models (Llama3.2-vision, etc.), the embeddings compatible with other modalities are also likely to be used in the future.
(2)	I would very much hope the authors to discuss impacts of fine-tuning on Prompt Detective. Given large number of fine-tuned (domain-specific) LLMs, and we know that fine-tuning would significantly change the response distributions. Therefore, this method would be less effective (large-bias may present) on fine-tuned LLMs (e.g. Medical-related GPTs).
(3)	The paper could benefit from a detailed analysis/presentation of computational costs in terms of query frequency, model access, and performance trade-offs. This would provide practical insights for implementing the method in commercial settings.
(4)	I think authors also need to discuss (a lot) more on the safety/privacy-related implications in discussion, especially what are some of recommendations or practical steps for ML/AI practitioners to protect their intellectual properties.

---

### Official Review · Reviewer_DDsG · 2024-10-09
**In this paper, the authors propose a membership inference approach for detecting proprietary system prompt usage in LLMs. They introduce Prompt Detective, a statistical method that compares the distributions of responses generated by LLMs to identify whether a particular system prompt has been used. Their experimental results show that Prompt Detective can reliably distinguish between prompts in challenging scenarios, including black-box settings, and when dealing with similar prompts.**

**Rating:** 6
**Confidence:** 4

**Review:**

Strengths

The idea of using membership inference to detect prompt usage of proprietary systems in LLMs is innovative. Most existing work focuses on training-based or model-specific detection techniques, but these approaches cannot handle black-box settings, or situations where prompt variations are minimal. This limits their applicability in real-world scenarios. This paper introduces a method that leverages statistical comparisons of response distributions, as well as improves the ability to detect and distinguish between similar prompts without requiring model retraining.

The workflow is well-structured, as it adopts a statistical analysis approach to systematically compare outputs to make the detection process more robust and adaptable to various LLMs. The authors eliminate the need for additional model training and further enhance the scalability of prompt detection across different models and deployment conditions.

The experiments are extensive, with detailed analysis of the results. These experiments validate the effectiveness of Prompt Detective and demonstrate the method's reliability and accuracy across different model architectures and levels of prompt similarity​.

Weaknesses

In Section 3, Methodology, the assumption that similar prompts can be detected solely through statistical distribution comparisons could oversimplify the complexities of LLM responses. The paper does not sufficiently address edge cases where two different prompts could generate highly similar outputs due to inherent biases in the language model.

Also, the paper does not provide sufficient comparisons with existing state-of-the-art techniques for prompt detection, such as adversarial testing or model-specific approaches (Section 4, Related Work).

Overall, the paper presents a promising approach with Prompt Detective for detecting proprietary prompt usage in LLMs, but it could benefit from more extensive comparisons with SOTA methods and a deeper exploration of prompt diversity.

---

### Official Review · Reviewer_aNkW · 2024-10-09
**Review for Has My System Prompt Been Used?**

**Rating:** 5
**Confidence:** 3

**Review:**

This paper presents a method, named Prompt Detective, for prompt membership inference aimed at detecting whether a proprietary system prompt has been used in a third-party language model. The authors measure the similarity of responses by calculating the distribution of response embeddings from both the suspected and known proprietary models. Based on cos similarity and statistical tests, the proposed system determines if the properitary prompt has been used based on whether the obtained responses are similar.

Quality: The proposed method is technically sound and validated through evaluations.
(Please see the weakness section for my concern.)

Clarity: The paper is clear in terms of the problem, methodology, and results.

Originality: The paper’s originality is limited due to its reliance on established techniques.
The application of this paper is on prompt membership inference.

Significance: The proposed method tackles the problem of prompt membership inference. The paper enhances prompt security by verifying whether a specific system prompt has been used.
(Please see the weakness section for my concern.)


Strenght:
1. Provides a structured approach for prompt membership inference detection.
2. conducted evaluations with different LLMs/scenarios, including standard and hard prompts (varying degrees of similarity)  and a black box setup.

Weakness:
1. Regarding making the decision based on results from the similarity, my concern is that this method may fall short when evaluating two LLMs that are trained/finetuned on distinctive domains or datasets (For example, general QA vs Medical, or science vs literacies). How to tell whether the difference in statistics is due to model difference or wheather the proprietary system prompt is used. Similarly, if two LLMs are trained on exact same datasets, their reponses may be similar.

2, Generation to more LLM models: Here the proposed method makes the decision based on a quantified measurement of the similarity of the responses using embedding like Bert.
For practical reasons, one can simply train on Bert's text-classification algorithm. How would the proposed method perform compared to this approach?